# Environmental and Food Safety Assessment of Pre-Harvest Activities in Local Small-Scale Fruit and Vegetable Farms in Northwest Portugal: Hazard Identification and Compliance with Good Agricultural Practices (GAPs)

**DOI:** 10.3390/foods14122129

**Published:** 2025-06-18

**Authors:** Ariana Macieira, Virgínia Cruz Fernandes, Teresa R. S. Brandão, Cristina Delerue-Matos, Paula Teixeira

**Affiliations:** 1CBQF—Centro de Biotecnologia e Química Fina—Laboratório Associado, Escola Superior de Biotecnologia, Universidade Católica Portuguesa, Rua Diogo Botelho 1327, 4169-005 Porto, Portugalpcteixeira@ucp.pt (P.T.); 2REQUIMTE/LAQV, Instituto Superior de Engenharia do Porto, Instituto Politécnico do Porto, Rua Dr. António Bernardino de Almeida, 431, 4249-015 Porto, Portugal; vircru@gmail.com (V.C.F.);; 3Ciências Químicas e das Biomoléculas, Escola Superior de Saúde, Instituto Politécnico do Porto, Rua Dr. António Bernardino de Almeida 400, 4200-072 Porto, Portugal

**Keywords:** fresh produce, pre-harvest, small-scale farming, microbiological contamination, chemical contamination

## Abstract

The popularity of small-scale and local fruit and vegetable production has increased in recent years due to perceived economic, environmental, and social benefits. However, these operations face contamination risks that both consumers and small-scale producers may underestimate. The present study aimed to assess the microbiological and chemical hazards on fruit, vegetables, soil, and water samples from small-scale farms in north-western Portugal during pre-harvest activities. Additionally, the study investigated farmers’ non-compliance with food safety regulations and good agricultural practices (GAPs), exploring how their behaviour might contribute to the identified hazards. A before-and-after analysis of non-compliant behaviours was conducted to determine the impact of training on improving food safety practices. The analysis identified the presence of pathogenic bacteria, pesticides, flame retardant residues, nitrates, and heavy metals. Lead (Pb) concentrations exceeded EU limits in organic carrots from one producer (0.156 ± 0.043 mg/kg) and in chard from another (0.450 ± 0.126 mg/kg). Cadmium (Cd) levels were also above regulatory thresholds in bell peppers (0.023 ± 0.009 mg/kg) and organic tomatoes (0.026 ± 0.015 mg/kg) from two different producers. Elevated levels of heavy metals were detected in irrigation water from two sites, with zinc (Zn) at 0.2503 ± 0.0075 mg/L and Pb at 0.0218 ± 0.0073 mg/L. Among food samples, the most prevalent microorganisms were *Pseudomonas* spp. (88.2%), *Bacillus cereus* (76.5%), and aerobic mesophilic bacteria (100%). Phosphorus flame retardants (PFRs), particularly tris(2-butoxyethyl) phosphate (TBEP), were detected in all food and soil samples. Some EU-banned pesticides were detected in food and soil samples, but at levels below the maximum residue limits (MRLs). Chlorpyrifos (35.3%) and p,p’-DDD (23.5%) were the most detected pesticides in food samples. After the training, GAP behaviour improved, particularly that related to hygiene. However, issues related to record-keeping and soil and water analyses persisted, indicating ongoing challenges in achieving full compliance.

## 1. Introduction

Local food production has grown in popularity [1] due to its perceived benefits, including improved taste, nutrition, and safety; support for local economies [2,3]; environmental benefits [4]; reduced carbon footprint [5]; improved animal welfare [6]; increased traceability [7]; and greater transparency for consumers [8]. In 2019, local food markets in the United States generated more than USD 20 billion in sales [9]. However, fruits and vegetables from local sources can still pose a health risk to consumers. This risk arises from various potential sources, which can be classified as microbiological, chemical, or both. Microbiological risks include contamination from wildlife and pets, soil and manure, irrigation water, and inadequate handling and distribution practices. Chemical risks primarily originate from agrochemicals and may also be linked to soil and irrigation water due to the presence of pesticide residues, heavy metals, and nitrates. Certain sources, such as soil and irrigation water, can pose both microbiological and chemical hazards [10]. Despite the claimed benefits of local food production, these risks of contamination highlight the need for continued vigilance in food safety practices at all levels of farming.

Pathogenic bacteria, including *Salmonella* spp., *Listeria monocytogenes*, *Escherichia coli* O157, *Staphylococcus aureus*, *Bacillus cereus*, and *Clostridium perfringens*, are a major food safety concern in various fruits and vegetables (e.g., lettuce, cucumbers carrots, tomatoes, apple juice) [11,12,13]. *E. coli* was found in 29% of lettuce samples at farmers’ markets in Pennsylvania [14]. *Listeria monocytogenes* has been detected in leafy greens and spinach in Florida [15], and in lettuce, spinach, and kale in Pennsylvania [14]. Assessing microbial safety hazards during pre-harvest activities is important [16], as contamination that occurs during production is often carried through to later stages and may not be fully removed by post-harvest interventions.

Zikankuba et al. [17] reported that 25% of all pesticides are used on vegetables, but 99.9% fail to reach their target and end up in the environment and food supply [18]. Common pesticides, including organochlorines (already banned), organophosphorus and pyrethroids, are toxic and have been linked to cancer, mutations, teratogenicity [19], bioaccumulation, and long-term persistence [20]. The European Commission (EC) set maximum residue limits (MRLs) for pesticides in all food products by Regulation (EC) No 396/2005 [21].

Persistent organic pollutants (POPs), including organochlorine pesticides (OCPs), polychlorinated biphenyls (PCBs), and brominated flame retardants (BFRs), are characterised by their high stability, bioaccumulation, and persistence in the environment [22,23]. The Stockholm Convention and EU Regulation 2019/1021 restrict POPs because of their serious health risks, including cancer and various disorders [24,25,26]. According to the UN Environment Programme [27], flame retardants are chemicals added to materials such as plastics, textiles, and paints to inhibit, suppress or delay flames and prevent fires. Flame retardants, such as brominated flame retardants (BFRs), have been widely used in industry and are considered POPs. Phosphorus flame retardants (PFRs) are not classified as POPs due to their lower persistence and bioaccumulation. However, some PFRs can still be harmful to the environment and human health because they persist in the aquatic environment [28]. PFRs have been used as alternatives to BFRs [29].

Nitrate (NO_3_^−^) is a chemical ion utilised by vegetables and fruits as a source of nitrogen. According to the European Food Safety Authority(EFSA) [30], 80% of human nitrate intake comes from raw vegetables. EU Regulation 2023/915 limits the concentration of nitrate in food [31]. Nitrate may be carcinogenic and has been associated with stomach and oesophageal cancer [32], methemoglobinemia [33], and mutagenic and teratogenic effects [34].

Heavy metals are dispersed in the environment through natural processes and human activities, such as sewage sludge application, fertilisers, incinerators, vehicle emissions, industry, and the presence of microplastics [35,36]. Vegetables can accumulate these metals, posing health risks, including cancer, and neurological, genetic, and reproductive disorders [37,38]. Among these, cadmium (Cd) contamination is of particular concern as it has been linked to prostate disease, chronic lung conditions, and kidney failure [37]. Similarly, lead (Pb) exposure also poses serious health risks, including mental impairment and reproductive problems [37]. Although Regulation (EU) No 2023/915 sets limits for Cd and Pb in food to reduce risks [31], studies show significant contamination of vegetables with these compounds [39,40].

Good agricultural practices (GAPs) encompass a set of principles, regulations, and technical guidelines aimed at ensuring the safe production of fresh fruits and vegetables [41,42]. Implementing GAPs and food safety rules plays a crucial role in preventing the introduction of pathogens and chemical contaminants into the fresh produce supply chain [43]. European regulation (EC) No 852/2004 should be followed by farmers as it establishes the hygiene requirements for foodstuffs.

The aim of this study was to assess microbiological and chemical hazards in fruit, vegetables, soil, and water samples collected from small farms in north-western Portugal during pre-harvest activities. Additionally, the study evaluated the level of non-compliance with food safety regulations and GAPs, among farmers, exploring how their practices could contribute to the identified hazards. A comparative analysis of farmer behaviour before and after training was also conducted to assess the effectiveness of the training in improving food safety practices.

## 2. Material and Methods

### 2.1. Sampling

Water, soil, and produce samples were collected from 17 local small-scale farms in the districts of Braga and Porto in northern Portugal. The locations and identification of these farms are detailed in Figure 1. A specific fruit or vegetable type was selected for analysis from each farm, focusing on varieties typically consumed fresh and thus posing a higher risk of foodborne illness. From each farm, one kilogram of produce was collected from various locations within the cultivation plots and then combined to create a composite sample. Similarly, one kilogram of soil was gathered from different spots within each plot using disinfected shovels in a zigzag pattern and then combined into a composite sample [44]. Additionally, two and a half litre samples of irrigation water were collected in sterile flasks from their respective sources. Samples intended for microbiological analyses were stored overnight at 4 °C, while those for chemical analyses were kept at −20 °C for a maximum of 10 days.

### 2.2. Microbiological Analysis

Twenty-five grams of each unwashed product was added to 225 mL of sterile buffered peptone water and homogenized in a Stomacher BagMixer (Interscience, Saint Nom la Brèteche, France) for 1 min at speed 2, which corresponds to a moderate homogenization setting as defined by the manufacturer. Decimal dilutions were prepared in Ringer’s solution for the enumeration of *E. coli* [45], *Listeria* spp. [46], *Pseudomonas* spp. [47], *B. cereus* [48], *C. perfringens* [49], coagulase-positive staphylococci [50], and the detection of *L. monocytogenes* [51] and *Salmonella* spp. [52]. Hygiene indicator microorganisms in food, such as Enterobacteriaceae [53], yeasts, and moulds [54] and aerobic mesophilic bacteria [55] have also been enumerated. The detection limits for *Listeria* spp., *Pseudomonas* spp., *B. cereus*, and coagulase-positive staphylococci were less than 2.3 CFU/g, rather than less than 2 CFU/g, due to the spreading method utilized with a Spiral equipment (easySpiral, Interscience, France). The same procedure was applied to soil samples for the enumeration and detection of pathogenic microorganisms. Since there are no specific ISO guidelines for enumerating and detecting pathogenic microorganisms in soil samples, ISO standards for food were used as a reference instead. ISO methods developed for food matrices were used as a reference due to their proven reliability and ability to handle complex biological samples. These methods target pathogens commonly found in both food and soil, offer high sensitivity and specificity, and can be adapted to accommodate the unique characteristics of soil. Using well-validated food microbiology protocols provides a practical and robust approach until soil-specific standardized methods become available.

Irrigation water was analysed using the membrane filtration technique for volumes of 100 mL and 10 mL. ISO standards were followed for the enumeration of *E. coli* [56], *Pseudomonas* spp. [57], and *C. perfringens* [58] and for the detection of *Salmonella* spp. [59]. The enumeration of *Listeria* spp., *B. cereus,* and coagulase-positive staphylococci was performed by membrane filtration using the same culture media as for food and soil samples. The detection of *L. monocytogenes* was performed using the same procedures as for food and soil samples, as there are no ISO guidelines for the enumeration and detection of this pathogen in water.

### 2.3. Chemical Analysis

#### 2.3.1. Pesticide, PCBs, and Flame Retardants Compounds

The list of pesticides, PCBs, and flame retardants analysed is given in the Appendix A (Appendix A). Standard agrochemical compounds for OPPs, OCPs, and pyrethroids were supplied by Sigma-Aldrich (99%) (St. Louis, MO, USA). POPs such as brominated flame retardants (BFRs; ≥98%) and polychlorinated biphenyls (PCBs) were supplied by Isostandards Material, S.L. (Madrid, Spain) at 50 mg/L in isooctane and Riedel-de Haën (Seelze, Germany) at 10 ng/μL in isooctane, respectively. Phosphorus flame retardants (PFRs; 99%) were supplied by Sigma-Aldrich (St. Louis, MO, USA). The extraction procedure used the original QuEChERS method (quick, easy, cheap, effective, rugged, and safe) with dispersive solid phase extraction (d-SPE) cleanup (with 150 mg MgSO4, 50 mg PSA, and 50 mg carbon) from Agilent Technologies (Santa Clara, CA, USA). The extraction procedure followed the protocols described in previous studies [60,61,62,63].

Gas chromatographic (GC) analysis was performed using two different detectors: an electron capture detector (ECD; Shimadzu GC-2010 gas chromatograph, Shimadzu, Kyoto, Japan), according to Fernandes et al. [61] and a flame photometric detector (FPD; Shimadzu GC-2010 Plus gas chromatograph, Shimadzu, Kyoto, Japan), according to Fernandes et al. [64]. Standard solutions were prepared to generate calibration curves.

#### 2.3.2. Nitrate

Sample preparation and extraction were conducted according to the method developed by Hongsibsong et al. [65] and Uddin et al. [34]. After extraction, the samples were introduced into the high-performance liquid chromatography (HPLC) system (Beckman Coulter, Brea, CA, USA) equipped with a photo diode array (PDA) detector (Beckman Coulter, System Gold 168, Brea, CA, USA). A C18 column was used for the separation of the compounds (Alltima Amino, 100 Å, 250 × 4.6 mm, 5 µm). The column oven temperature was 40 °C, and the running time was 10 min. The retention time for nitrate was 3.8 min. Standard solutions were prepared to generate calibration curves.

#### 2.3.3. Heavy Metals

Samples were processed using a microwave acid digestion system (MARS 6, CEM Corporation, Matthews, NC, USA) as per standard protocols. A positive sample (with the added metal) and a negative control (blank) were also included in the procedure. The equipment operated according to the parameters outline in the MARS 6, Microwave Acid Digestion, Compendium, specifically tailored for soil and food samples.

Following microwave digestion, the sample products were cooled to room temperature, then transferred to 50 mL tubes, and diluted with deionized water (Milli-Q™ Model 3UV ELIX, Millipore Corporation, Burlington, MA, USA) to reach the same volume. For water samples, 1% HNO_3_ was added to a final volume of 15 mL to aid in sample digestion.

The elements assessed in this study included cadmium (Cd), zinc (Zn), lead (Pb), nickel (Ni), copper (Cu), chromium (Cr), and mercury (Hg). The primary instrument used for quantifying most heavy metals was the ICP-OES (ICP-OES/optima 7000DV; Perkin Elmer, Waltham, MA, USA). However, due to the inherent instability of Hg, using ICP is not a suitable method for its detection. Therefore, it was measured using the atomic absorption spectophotometry cold vapour (AASCV) system (Perkin Elmer, Analyst 600, Waltham, MA, USA). For both methods, standard solutions were prepared to generate calibration curves for each element.

### 2.4. Evaluation of Farmers’ Behaviour Regarding GAPs Before and After Training

Simultaneously with microbiological and chemical analyses, farmers were surveyed regarding their compliance with GAPs. Based on the results from both the microbiological and chemical analyses and the farmers’ responses, a series of training sessions focusing on GAPs and food safety were developed and implemented to enhance farmers’ practices and align them with GAPs and food safety regulations in pre-harvest activities. Six months later, farmers were asked the same questions as before the training, and their responses were compared to the initial data.

The survey questions were directly related to compliance with GAPs and were presented in a yes/no format using the present simple tense. The sequence of these questions matched the responses later presented in Figure 4 (Results and Discussion section). Examples included the following: “Do you know the surrounding areas of the production site?” and “Is your production planned and organized?”

### 2.5. Data Analysis

Three replicates were carried out for each of the materials analysed, including both chemical and microbiological indicators.

A principal component analysis (PCA) was performed using the Varimax method. This analysis included all the parameters studied, except for four variables: *Enterobacteriaceae*, yeasts, moulds, and aerobic mesophilic bacteria counts. These variables were ex-cluded from the PCA as they were not assessed in the soil due to their naturally elevated levels in such samples. Variables were included in principal components if their loadings exceeded 0.5 in the respective component loadings.

The Mann–Whitney test was used to assess the differences between the raw material and water samples regarding the number of Enterobacteriaceae, yeasts, moulds, and aerobic mesophilic bacteria, after normality as assessed by the Shapiro–Wilk test had not been confirmed. A 5% significance level was used for all analyses.

Data analysis was performed using IBM^®^ SPSS^®^ Statistics (version 28), Microsoft Excel and a python data visualization library named Matplotib v3.10.1.

## 3. Results and Discussion

### 3.1. Assessment of Microbiological and Chemical Hazards

A PCA was performed to identify the main factors contributing to the variation in the farmers’ samples. Variables falling between the limits of detection (LODs) and the limits of quantification (LOQs) for chemical analysis and those above the LODs for microbiological analysis were included in the PCA design. The covariance between PC1 and PC2 was found to be zero. The first component (PC1) explained 32.17% of the variance, while the second component (PC2) explained 10.00%. In PC1, heavy metals, coagulase positive staphylococci, *Listeria* spp., *E. coli* and *C. perfringens* were positively loaded and contributed significantly to the variation explained by this component. The compound 2,2′,4,4′,5,5′-hexachlorobiphenyl (PCB153), the pyrethroid bifenthrin, specific PFR compounds such as tris(2-ethylhexyl) phosphate (TEHP) and tris (2-butoxyehyl) phosphate (TBEP), and bacterial species such as *B. cereus* and *Pseudomonas* spp. showed positive values on PC2. The PCA score plot (Figure 2A) showed three distinct clusters corresponding to the food, soil, and water samples. Soil samples were predominantly aligned with the positive axis of PC1, while food samples showed a stronger association with the positive axis of PC2. Conversely, for water samples, features such as isomer tri-cresyl phosphate III isomer (TCPIII) and L. monocytogenes were identified as key contributors to the observed variance. These features had negative scores on both PC1 and PC2. In Figure 2A, the size of the circles reflects the level of *B. cereus* in the samples. Overall, soil samples had the highest levels, followed by food and water.

The clusters in Figure 2A showed a proximity between samples of the same type from all farms, indicating similar microbiological and chemical profiles within each sample category. NP, OP, and FP were farms from Porto district, with NP and OP from Paredes and FP from Felgueiras. These three were the farms that showed a significant influence of heavy metals and bacterial levels in the soil samples, as evidenced by their high positive scores on PC1 (Figure 2A). Such a situation could be attributed to similarities in soil characteristics and climatic conditions between nearby areas [66]. Hazards such as heavy metals, except Hg, should be considered as a potential concern, especially regarding soil samples. According to APA [67], none of the soil samples exceeded the established threshold for heavy metals (Table 1). However, it is important to control and analyse heavy metal levels because they can significantly impact soil safety. Although heavy metal contamination in the soil was not a concern, according to the Regulation (EU) N° 2023/915, the established limit for Pb was exceeded in samples of organic carrots from the LP producer and in samples of chard for raw salads from the OP farm (0.156 ± 0.043 and 0.450 ± 0.126 mg/kg of fresh weight, respectively) [31] (Appendix A). Cd concentrations (0.023 ±0.009 and 0.026 ± 0.015 mg/kg of fresh weight, respectively) were found in CB products (bell pepper) and MP products (organic tomatoes) above the regulatory limit (Appendix A). Only two producers detected heavy metals above the LOQ in their irrigation water (Table 1 and Appendix A). Water samples from producer EB had a Zn concentration of 0.2503 ± 0.0075 mg/L and those from farm PB had a Pb concentration of 0.0218 ± 0.0073 mg/L (Appendix A). According to the Portuguese law DL 236/98 [68], neither concentration exceeded the RMV. In line with soil results, food and water generally did not pose a risk to consumers in terms of heavy metal concentrations, as most samples were below the LOD or LOQ for all analysed elements (Table 1, Appendix A). Hg was the only element not detected in any sample type (food, soil or water) (Table 1).

Together with the heavy metals, bacteria such as coagulase-positive staphylococci, *Listeria* spp., *E. coli*., and *C. perfringens* were identified as factors contributing to variance among soil samples. Vegetables from farms HP, OP, LP, and KP, which were grown closer to the soil, showed more problems regarding microbiological indicators. *Listeria* spp., coagulase-positive staphylococci and *C. perfringens* were present in all the soil samples (Table 2). Soil can be a relevant factor for food contamination, particularly for foods in direct contact with it [69]. *Enterobacteriaceae*, aerobic mesophilic bacteria, yeasts, and moulds were not tested in soil samples due to their naturally high levels, which result from environmental contamination and agricultural practices such as fertilisation [70,71,72].

Concerning the food samples, organic carrots from LP appeared to be the primary sample affected by chemical compounds such as TEHP, PCB153, and bifenthrin. As previously noted, soils can serve as significant sources of contamination during pre-harvest activities [73]. Given that carrots are root vegetables that grow within the soil, they may be more susceptible to contamination. The remaining food samples could be grouped together, influenced by factors such as the level of *B. cereus*, *Pseudomonas* spp., and TBEP.

*Pseudomonas* spp. (88.2%) and *B. cereus* (76.5%), along with aerobic mesophilic bacteria (100%), were the most common microorganism found in food samples (Table 2). *Pseudomonas* spp. and aerobic mesophilic bacteria are commonly used as indicators of food quality and hygiene, while *B. cereus* is recognized as a foodborne pathogen [74,75]. According to Tango et al. [76], *B. cereus* showed the highest prevalence in fresh vegetables and fruits in Korea (50.3%). According to the same author, some studies reported high contamination levels in vegetables, with a prevalence of 70% prevalence in South Korea and 57% in Mexico City. This widespread occurrence may be due to the ubiquitous distribution of *B. cereus* in nature [77]. The levels of aerobic mesophilic bacteria in vegetables were consistent with those reported by Tango et al. [76].

*Salmonella* spp. and *L. monocytogenes* were not detected in any of the food or soil samples (Table 2). However, they are commonly found in pre-harvest vegetables and fruits and have been associated with foodborne outbreaks linked to these types of commodities [12,38,78,79].

In our study, higher levels of *Enterobacteriaceae* were observed in lettuce from farm HP and tomatoes from farm PB. In contrast, *Enterobacteriaceae* levels were below the LOD in all fruits analysed (Table 2). These results are in line with the research of Leff and Fierer [80], who reported that lettuce and tomatoes, among other vegetables, showed higher relative abundances of members of the *Enterobacteriaceae* family compared to fruits like apples.

A distinct cluster was also observed in the water samples. FP and GP, both from Felgueiras and OP from Paredes, showed similar characteristics regarding the safety of their irrigation water (Figure 2A). These producers seemed to be more affected by PFRs such as TCP III and *L. monocytogenes* contamination compared to the other farmers.

Although *Pseudomonas* spp. and aerobic mesophilic bacteria were not identified as relevant contamination factors according to PCA, they were present in all water samples (Table 2). Irrigation water of farms DB, GP, HP, JB, and OP had the highest levels of microorganisms, with the additional concern that *L. monocytogenes* was detected in the water from GP (Table 2). These water samples could therefore pose a greater risk of contamination of soil and pre-harvested produce, particularly for HP and OP, where produce such as lettuce and chard were grown directly in the soil.

The results of the Mann–Whitney test showed no significant differences between food and water samples for the levels of moulds (*p* = 0.085), yeasts (*p* = 0.812), and *Enterobacteriaceae* (*p* = 0.099). However, significant differences were observed for aerobic mesophilic bacteria (*p* < 0.001), with higher levels detected in food samples compared to water samples (Figure 2B).

PFRs were important contributing factors, mainly for food and water contamination, as shown in PCA. TBEP was quantified in all the food and soil samples, making it the most quantified PFR compound in this study, followed by TEHP in food (23.5%), tri-cresyl phosphate I isomer (TCP I), and tri-n-butyl phosphate (TnBP) in soil (5.9% for both) (Figure 3C and Table 1).

PFRs were detected in all farms’ food and soil samples, with TBEP, TEHP, TCPs, and TnBP being the most prevalent (Figure 3A and Table 1). The high concentrations of PFR compounds observed in this study, might have been due to environmental contamination. These compounds are extensively utilized in industrial applications, often serving as substitutes for banned BFRs [81,82]. The industrial substitution of BFRs by PFRs could also explain why no BFR compounds were detected in any of the samples in this study, and why PFRs were more frequently detected in the environment than BFRs [83]. TCP III was detected above the LOQ in the irrigation water samples from farms LP and OP. The low frequency of PFR contamination in water samples might be due to the use of headwaters for irrigation (Figure 1). TBEP was found above the LOQ in all food and soil samples and was the most frequently detected PFR in this study (Table 1 and Appendix A). As PFRs can be taken up by plants [82,83], soil contamination could lead to food contamination not only through direct contact but also through plant uptake. PFRs were mainly found in vegetables, which is in line with the results of the study by He et al. [81]. While replacing BFRs with PFRs may contribute to the reduction of POPs in the environment, it is also important to consider the potential increase in PFR compounds and their associated effects.

Bifenthrin was the only pesticide residue considered as an important factor for food contamination, mainly in LP samples, although several pesticides and flame retardants were found to exceed their LOQ, especially in food and soil samples (Table 1 and Appendix A). Results falling between the LOD and LOQ were considered in this study, following the approach outlined in the EFSA [84] report.

Agrochemicals such as chlorpyrifos were quantified in 35.3% of the food samples, followed by 1-chloro-4-[2,2-dichloro-1-(4-chlorophenyl) ethyl] benzene (p,p’-DDD) and bifenthrin (23.5 and 5.9%, respectively; Figure 3D and Table 1). p,p’-DDD was the second most frequently quantified pesticide in food and soil samples (23.5 and 17.6%, respectively), but it was the most frequently detected in food samples (Figure 3B,D and Table 1). Chlorpyrifos was the most frequently quantified residue in water samples (35.3%), followed by bifenthrin (11.8%; Figure 3D and Table 1). Additionally, chlorpyrifos was the most frequently quantified residue in soil and the second most frequently quantified in food samples (Figure 3D and Table 1).

Several agrochemical residues were detected in 52.9% and 64.7% of the farmers’ food and soil samples, respectively. However, only 17.6% of the farms showed the combination of more than one quantifiable residue (above the LOQs) in both food and soil samples (Table 1 and Appendix A). In water samples, only farm KP had multiple residue contamination with concentrations of chlorpyrifos and bifenthrin (0.016 ± 0.000 and 0.213 ± 0.000 µg/kg) above the LOQs (Appendix A). Among the organic farms, food and soil samples from AB showed contamination with multiple agrochemicals; food samples from LP organic farms showed residues of bifenthrin above the LOQs (3.12 ± 1.53 µg/kg) (Table 1 and Appendix A) together with the detection of p,p’-DDD residues. In addition, chlorpyrifos was quantified in soil samples from the LP farmer (0.13 ± 0.00 µg/kg; Appendix A). Chlorpyrifos was quantified in irrigation water samples from farms AB and LP (Table 1).

Pesticide residues such as chlorpyrifos-methyl and dimethoate were not quantifiable but were detected in food, soil, and water samples (Figure 3B and Table 1). Both pesticides have been banned by the EU, the first in 2020 and the second in 2019. However, none of the agrochemical residues detected exceeded the MRLs set by the European Commission [85,86]. Chlorpyrifos and p,p’-DDD were the most abundant pesticides in food and soil samples (above the LOQs) (Figure 3D and Table 1 and Appendix A). Both pesticides have been banned in the EU [85,86]. The first was banned in 2020 due to concerns about its adverse effects on human health, and the second due to its high toxicity and persistence in the environment over long periods. p,p’-DDD, although not considered a residue itself, is a breakdown product of 1,1,1 trichloro-2,2-bis-(p-chlorophenyl) ethane (o,p’-DDT). Research by Shen et al. [87] showed that p,p’-DDD was the predominant compound among the o,p’-DDTs detected in all vegetable samples analysed. Chlorpyrifos has been one of the most frequently detected pesticides in vegetables and fruits [64,88,89]. Bhandari et al. [90] concluded that chlorpyrifos and p,p’-DDD were the most prevalent pesticides in soil samples. These findings align with the results of our study and confirm the prevalence of these pesticides in agricultural environments. Nearly half of the farmers had quantifiable residues of banned pesticides in their water samples (41.18%). Bifenthrin and chlorpyrifos were quantified in low concentrations in the irrigation water, while chlorpyrifos-methyl and p,p’-DDD were detected (below the LOQ) in 35.29% and 17.6% of the water samples, respectively (Figure 3B,D and Table 1 and Appendix A). The presence of different types of pesticides in irrigation water may contribute to indirect contamination of soil and food. Pesticides can cycle through water systems, contaminating not only surface water but also leaching into groundwater, which is often used for irrigation purposes [91].

The quantification of OCPs such as o,p’-DDTs, aldrin, and endosulfan I in the environment may be attributed to their high capacity for environmental persistence [61].

Nitrate concentrations in food, soil, and water samples are shown in Appendix A and analytical performance results are provided in Appendix A.

Farmers growing bok choy (KP sample), carrots (LP sample), chard (OP sample), and lettuce (HP sample) with two tomato producers (DB and PB samples) had elevated nitrate levels compared to the mean concentration found by EFSA [30] (Table 3). However, it is worth noting that the nitrate concentration in lettuce was below the maximum level set by Regulation (EU) N° 2023/915 [31] (Appendix A). Most of the soil samples exhibited nitrate levels surpassing the threshold limit set by the Department of Employment, Economic Development and Innovation in Queensland, Australia [92] (50 mg/kg, Table 3) (Department of Employment, Economic Development and Innovation, 2010), indicating the potential for nitrate leaching into groundwater, posing risks to both human health and the environment [93,94,95,96]. The irrigation water from farms BB and JB exceeded the maximum recommended value (MRV) of 50 mg/L for nitrate as set out in DL 236/98 [68] and in the Directive 2006/118/EC [97], suggesting that these water sources may contribute to nitrate contamination of both soil and food (Table 3).

### 3.2. Impact of Training on Farmers’ Adherence to Good Agricultural Practices

Microbiological contamination, primarily caused by *Enterobacteriaceae* and *E. coli* detected on the farms, may have been linked to certain non-compliant behaviour among farmers. These included the use of immature manure that had not undergone the required thermal processing at the ideal time and temperature before being applied to the soil (62% of farmers), and the absence of fertilization records and a fertilization plan (100% of farmers); simultaneously, due to the lack of microbiological and chemical soil analyses (conducted by only 5% of farmers), soil contamination remained undetected, and no corrective actions were implemented [98,99,100,101] (Figure 4). Poor personal and environmental hygiene practices (76% and 57% of farmers, respectively), along with the presence of wild and domestic animals in the production area (100% of the farmers), may have also contributed to microbiological contamination of produce, soil, and water [13,16,78,102,103] (Figure 4). Water contamination may be attributed to the fact that none of the farmers in this study conducted microbiological or chemical analyses of irrigation water, making potential contamination an unrecognized issue. Additionally, the practice of storing water in outdoor and/or open reservoirs further facilitates environmental and animal contamination [104,105], (Figure 1 and Figure 4). Most farmers irrigated their produce using a hose (Figure 1). Hose irrigation poses a significant risk due to its uncontrolled water flow, which can increase the likelihood of large volumes of water directly contacting the edible portions of vegetables [106]. Although washing vegetables and fruits is typically a post-harvest activity, 57% of the farmers reported using pre-harvest water to wash produce. Additionally, none of the producers kept records of raw materials sourced from suppliers, such as plants or seeds, suggesting that microbiological contamination could also originate from these raw materials.

Low levels of heavy metal contamination in soil and food may be explained by the fact that none of the farmers used wastewater or sewage sludge for agricultural purposes, and their fields were located in rural and remote areas with no nearby industry or heavily trafficked roads [107,108]. Satisfactory results for irrigation water samples could be attributed to the distance of the producers’ fields from industrial effluents and to the use of headwaters for irrigation [109]. For these reasons, Hg was not detected in any of the samples (food, soil, and water) from the farms.

Non-compliant food samples for Cd and Pb concentrations, according to Regulation (EU) N° 2023/915 [31], may be attributed to the use of seeds or plants already contaminated with these elements, as none of the farmers tracked their seed/plant suppliers or other relevant information regarding raw materials (Figure 4). Copper (Cu) was detected above the LOQ in a significant proportion of both food and soil samples (Table 1). This contamination could be linked not only to soil fertilization activities but also to the widespread use of Bordeaux mixture by farmers, primarily for treating mildew in plants [110,111]. Such contamination could potentially be mitigated if producers performed chemical soil analyses, which could lead to optimized application of Bordeaux mixture on crops, thereby reducing the risk of contamination from its use (Figure 4).

PFRs are known for their toxicity, ability to bioaccumulate, and persistence in the environment. The widespread use of plastic mulch on 67% of farms and the production of food in greenhouses by an equivalent percentage of farms (as shown in Figure 4) could also contribute to the presence of PFRs in the environment. According to Sun et al. [112] the concentration of BFR compounds tends to be higher in greenhouse vegetables than in field vegetables. Although the study by Sun et al. [112] focused on BFRs, this finding may help explain why 47.1% of the farms in this study (where food samples were taken from greenhouses) could have problems with PFR compounds. As shown in Figure 4, 67% of the farms also had various other plastic materials spread throughout their fields (Figure 4).

In this study, pesticide concentrations were slightly higher in food samples than in soil samples. Contamination of food can occur from soil or other environmental sources such as air or during pesticide application [60,113,114,115]. Geissen et al. [114] found that all soils analysed under conventional agriculture (including Portuguese soils) contained detectable levels of agrochemical residues. Environmental contamination (e.g., application by neighbours) can occur due to the high volatility and persistence of certain compounds over time [116]. This highlights the importance of understanding the surrounding areas of the production site, as reported by 48% of the farmers. Ineffective pesticide application strategies, poor record-keeping practices [117], and the use of outdated or EU-banned pesticides also contribute to potential risks. None of the farmers documented pesticide applications and pest and disease control measures, and 29% of them faced challenges in properly disposing of these compounds and their packaging [118] (Figure 4).

High nitrate levels in soil may be due to the use of commercial fertilizers in some cases (which have high concentration of ready-to-use nitrogen), as seen with farms CB, FP, HP, or compost, as used by farm OP [94,96,119,120] (Figure 1). The absence of a structured fertilization plan, the lack of information regarding raw materials, and the absence of fertilization records from all the farmers in this study, coupled with the lack of soil analysis in 95% of cases (Figure 4), may contribute to the high nitrate levels detected [121]. This study also highlights the importance of chemical water analysis, which was not conducted by any of the farmers, as required by DL 236/98 [68], to identify potentially hazardous nitrate concentrations.

As 90% of the farmers had never participated in a training program on food safety and GAPs, some of the results observed in this study, particularly those related to microbiological contamination and the presence of hazardous chemical compounds on their farms, could be linked to a lack of compliance with GAPs. However, after the training sessions, only 33% of farmers remained unaware of GAPs, as they did not attend the sessions specifically organized to address the issues highlighted in the previous results.

As shown in Figure 4, most of the non-compliant practices studied improved after the training sessions. The only two practices that remained unchanged in terms of frequency among farmers were the lack of control over animal entry into the production area and the absence of microbiological and chemical analyses of the soil. Although improvements were observed in practices related to microbiological and chemical analyses of water and record-keeping, these issues remained prevalent among most farmers (Figure 4). The ongoing challenges may be attributed to financial constraints faced by small-scale producers [122,123], as these analyses incur costs. Additionally, record-keeping may be difficult to implement, as many farmers operate single-person supply chains and often have full-time jobs outside of agriculture, which could hinder traceability [124,125].

After the training, farmers demonstrated improved hygiene practices [126], such as refraining from washing vegetables with irrigation water. They also became more aware of their surrounding environment, began removing plastics from the farms, and implemented thermal treatment for organic fertilizers, as illustrated in Figure 4. These findings align with those of Schmit et al. [127] and Olayemi et al. [128], who observed that farmers can improve their practices following training in food safety and GAPs.

**Figure 4 foods-14-02129-f004:**
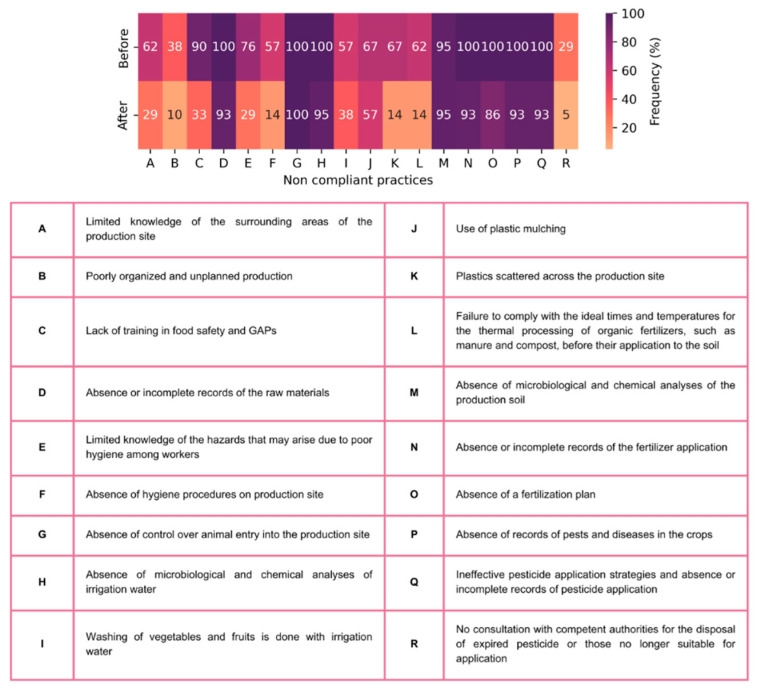
Frequency of farmers non-compliant to GAP standards before and after food safety and GAP training program.

## 4. Conclusions

This study found that microbiological and chemical hazards varied across farms, while certain parameters indicated the presence of hazards. Although only a small number of food samples contained heavy metals such as Pb and Cd at levels exceeding regulatory thresholds, these occurrences highlight the importance of further research into the associated health risks for consumers. The detection of agrochemicals banned by the EU in nearly all food and soil samples underscores the urgent need to improve local agricultural practices to mitigate such risks in fruit and vegetable production within short supply chains.

Although improvements were observed after training farmers in food safety and GAPs, further actions are needed to enhance compliance, particularly in traceability through record-keeping and facilitating microbiological and chemical analyses of soil and water, as these can be sources of contamination for fruits and vegetables from small-scale farms. To address these challenges, governmental support, combined with ongoing assistance from local development organizations, could help small farmers overcome these barriers. This support should not only provide financial aid but also promote awareness programs on potential hazards in agricultural production and encourage continuous collaboration with farmers to develop effective solutions.

Future research should aim to overcome the limitations of the current study, particularly the small sample size, in order to enhance statistical power and generalizability.

## Figures and Tables

**Figure 1 foods-14-02129-f001:**
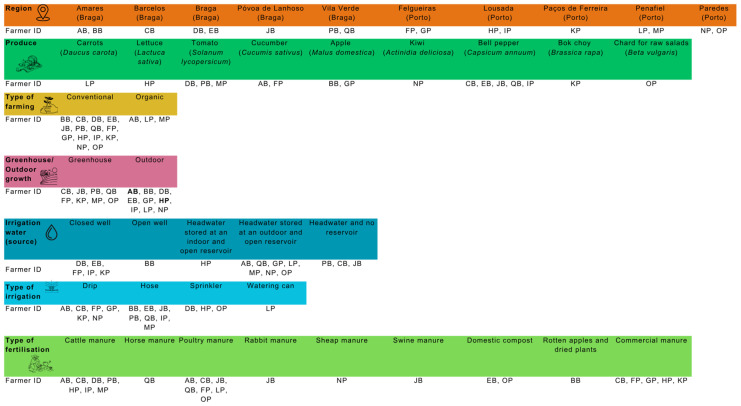
Farm locations, types of fruit and vegetables assessed in this study, and their environmental growing conditions (bold farmer IDs indicate that food samples were collected outdoors, but the farmers had greenhouses on their farms).

**Figure 2 foods-14-02129-f002:**
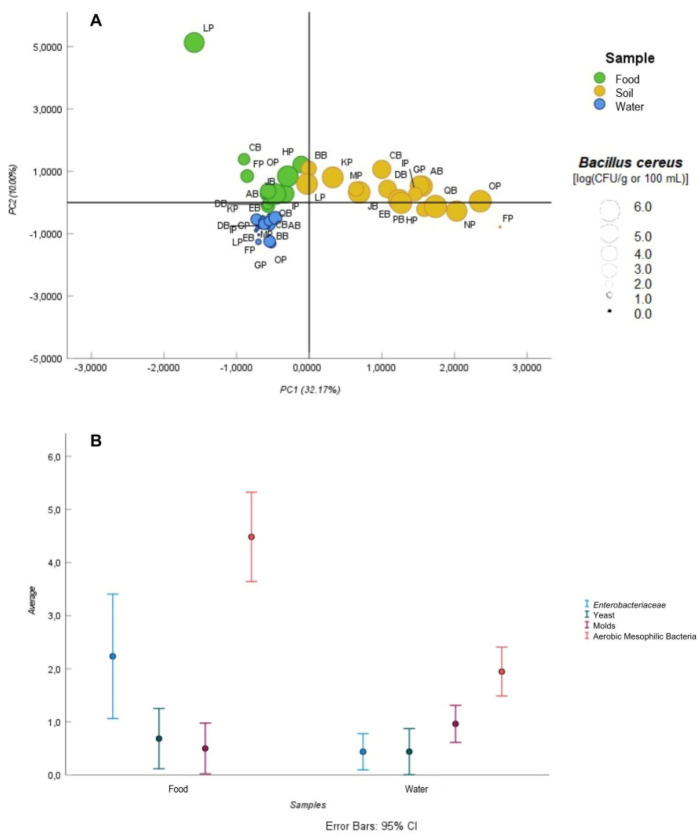
PCA score plot integrating a *B. cereus* quantification (**A**) and an average bar plot for Enterobacteriaceae, yeasts, moulds, and aerobic mesophilic bacteria with a 95% confidence interval (**B**).

**Figure 3 foods-14-02129-f003:**
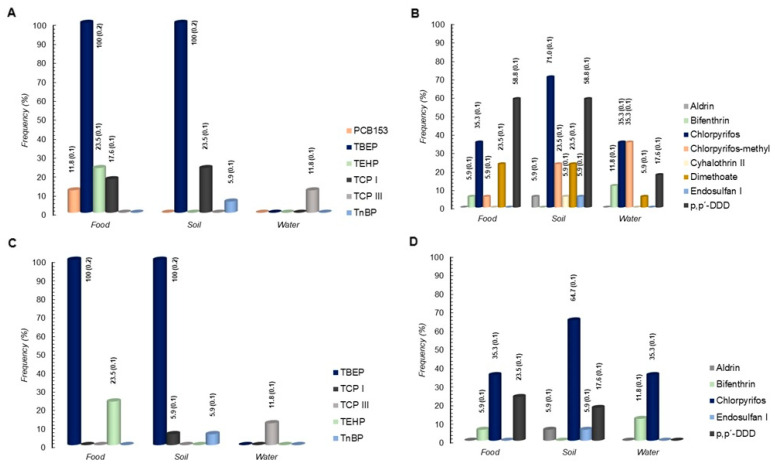
Detection (**A**,**B**) and quantification (**C**,**D**) of POPs/PFRs and agrochemicals compounds in food, soil, and water samples and their frequencies. Frequencies are followed by their respective standard errors in parentheses.

**Table 1 foods-14-02129-t001:** Percentage of food, soil, and water samples testing positive for chemical parameters (bolded samples indicate values above the quantification limits; bolded frequencies correspond to quantifiable samples, while non-bolded frequencies refer to samples where values were detected but not quantified).

	Food (Farmers ID)	% of Positives	Soil (Farmers ID)	% of Positives	Water (Farmers ID)	% of Positives
Heavy metals						
Zn	**AB**, BB, CB, **DB**, EB, **FP**, **HP**, JB, **KP, LP**, MP, **OP**, PB, QB,	82.4	AB, CB, DB, EB, **FP**, GP, HP, IP, JB, MP, NP, OP, PB, QB	82.4	**EB**	5.9
Cd	AB, BB, **CB**, **DB**, **EB**, FP, GP, HP, JB, **KP**, **LP**, **MP**, NP, **OP**, **PB**, **QB**	94.1	**AB,** BB, **CB**, **DB**, **EB**, **FP**, **GP**, **HP**, **IP**, **JB,** KP, LP, **MP**, **NP**, **OP,** PB, **QB**	100	JB	5.9
Pb	AB, BB, EB, HP, IP, **KP**, **LP**, MP **OP**	52.9	**AB**, BB, **CB**, **DB**, **EB**, **FP**, **GP**, **HP**, **IP**, **JB**, **KP**, LP, **MP**, **NP**, **OP**, **PB**, **QB**	100	CB, **DB,** FP, JB, HP, KP, LP, MP, OP	52.9
Ni	IP, **JB,** QB, **KP, LP, OP,** AB, FP, NP, HP, HP, PB	70.6	**AB, CB**, **DB**, **EB**, **FP**, **GP**, **HP**, **IP**, **JB**, **KP, MP**, **NP**, **OP**, **PB**, **QB**	88.2	NP	5.9
Cu	**BB, CB, EB, IP, JB, QB,** KP, **LP, AB, FP, NP, HP, DB, MP, PB**	88.2	**AB, CB**, **DB**, **EB**, **FP**, **GP**, **HP**, **IP**, **JB, MP**, **NP**, **OP**, **PB**, **QB**	82.4		
Cr	LP, **OP**	11.8	**AB, CB**, **DB**, **EB**, **FP**, **GP**, **HP**, **IP**, **JB**, **KP, MP**, **NP**, **OP**, **PB**, **QB**	88.2		
PCBs and flame retardant compounds						
PCB153	LP	5.9				
TBEP	**AB**, **BB**,**CB**, **DB**, **EB**, **FP**, **GP**, **HP**, **IP**, **JB**, **KP**, **LP**, **MP**, **NP**, **OP**, **PB**, **QB**	**100**	**AB**, **BB**, **CB**, **DB**, **EB**, **FP**, **GP**, **HP**, **IP**, **JB**, **KP**, **LP**, **MP**, **NP**, **OP**, **PB**, **QB**	**100**	---	---
TCPI	CB, LP, OP	17.6	AB, **GP**, KP, MP	**5.9**/ 23.5		
TCPIII					**LP**, **OP**	11.8
TEHP	**CB**, **LP**, **FP**, **HP**	**23.5**				
TnBP			**KP**	**5.9**		
Pesticide residues						
Aldrin			**CB**, KP	**5.9**/5.9		
Bifenthrin	**LP**	**5.9**			**JB**, **KP**	**11.8**
Chlorpyrifos	**GP**, **CB**, **IP**, **JB**, **FP**, **HP**	**35.3**	**AB**, **CB**, **DB**, **GP**, **IP**, **JB**, **KP**, **LP**, **OP**, **PB**, QB	**58.8/** 5.9	**AB**, **BB**, **GP**, **KP**, **LP**, **NP**	**35.3**
Chlorpyrifos-methyl	CB, AB, NP, EB	23.5	BB, CB IP	17.6	AB, BB, CB, HP, LP, QB	35.3
Cyhalothrin II			LP	5.9		
p,p’-DDD	**GP**, CB, **JB**, QB, LP, AB, **FP**, NP, DB, **PB**	**23.5/** 35.3	**AB, BB,** DB, IP, JB, LP, MP, **NP,** OP, PB	**17.6/** 41.2	EB, GP, PB	17.6
Dimethoate	CB, IP, JB, PB	23.5	BB, CB, DB, IP	23.5	EB	5.9
Endosulfan I			**BB**	**5.9**		

**Table 2 foods-14-02129-t002:** Percentage of food, soil, and water samples testing positive for microbiological parameters.

	Food (Farmers ID)	% of Positives	Soil (Farmers ID)	% of Positives	Water (Farmers ID)	% of Positives
*E. coli*	IP, HP	11.8	AB, EB, FP, GP, HP, NP, OP, PB, QB	52.9	GB, HP, JB, OP	23.5
*Listeria* spp.	IP, HP, LP	17.6	AB, BB, CB, DB, EB, FP, GP, HP, IP, JB, KP, LP, MP, NP, OP, PB, QB	100	AB, CB, DB, JB, LP, OP, PB, QB	47.1
*B. cereus*	AB, BB, CB, DB, FP, HP, IP, JB, KP, LP, NP, OP, QB	76.5	AB, BB, CB, DB, EB, GP, HP, IP, JB KP, LP, MP, NP, OP, PB, QB	94.1	AB, BB, CB, DB, EB, FP, GP, HP, IP, JB KP, MP, NP, OP, PB, QB	94.1
*Pseudomonas* spp.	AB, BB, CB, DB, EB, FP, HP, IP, JB, KP, LP, NP, OP, PB, QB	88.2	AB, BB, CB, DB, EB, FP, GP, HP, IP, JB, KP, LP, MP, NP, OP, PB, QB	100	AB, BB, CB, DB, EB, FP, GP, HP, IP, JB KP, LP, MP, NP, OP, PB, QB	100
Coagulase-positive Staphylococci	HP	5.9	AB, BB, CB, DB, EB, FP, GP, HP, IP, JB, KP, LP, MP, NP, OP, PB, QB	100	QB	5.9
*C. perfringens*	LP, OP	11.8	AB, BB, CB, DB, EB, FP, GP, HP, IP, JB, KP, LP, MP, NP, OP, PB, QB	100	AB, BB, DB, EB, FP, GP, HP, JB, LP, MP, NP, OP, PB, QB	82.4
*Enterobacteriaceae*	DB, HP, IP, JB, KP, LP, OP, PB, QB	52.9	---	---	AB, DB, GP, HP, JB, OP	35.3
Yeasts	HP, IP, JB, KP, OP	29.4	---	---	CB, JB, MP, PB	23.5
Moulds	CB, IP, KP, HP	23.5	---	---	BB, CB, DB, EB, GP, HP, IP, KP, LP, MP, NP, PB, QB	76.5
Aerobic mesophilic bacteria	AB, BB, CB, DB, EB, FP, GP, HP, IP, JB, KP, LP, MP, NP, OP, PB, QB	100	---	---	AB, BB, CB, DB, EB, FP, GP, HP, IP, JB KP, LP, MP, NP, OP, PB, QB	100
Detection of *L. monocytogenes*	---	---	---	---	FP, GP	11.8

**Table 3 foods-14-02129-t003:** Percentage of food, soil, and water samples above the established limits for nitrate.

Food ^a^ (Farmers ID)	% of Positives	Soil ^b^ (Farmers ID)	% of Positives	Water ^c^ (Farmers ID)	% of Positives
DB, HP, KP, LP, OP, PB	35.3	AB, BB, CB, DB, EB, FP, GP, HP, IP, JB, KP, MP, NP, OP, QB	88.2	BB, JB	11.8

a EFSA [30]. b Department of Employment, Economic Development and Innovation in Queensland, Australia (50 mg/kg) [92]. c DL 236/98 [68] and and Directive 2006/118/EC [97].

## Data Availability

The original contributions presented in the study are included in the article/Appendix A, further inquiries can be directed to the corresponding author.

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
