# Peer review of "Environmental and Food Safety Assessment of Pre-Harvest Activities in Local Small-Scale Fruit and Vegetable Farms in Northwest Portugal: Hazard Identification and Compliance with Good Agricultural Practices (GAPs)"

_foods, 2025, doi:10.3390/foods14122129_

Round 1

Reviewer 1 Report

Comments and Suggestions for Authors

This study systematically evaluates microbial and chemical contamination profiles in agricultural ecosystems by thoroughly analyzing produce, soil, and water samples collected from small-scale farming operations. The findings highlight the positive impact of targeted training programs on improving adherence to food safety protocols, providing empirical evidence to support risk-based management strategies for limited-resource farming systems. While the investigation offers significant insights, two critical limitations in the experimental design should be noted. First, the small sample size may restrict statistical power, potentially reducing the generalizability of the findings across different agricultural contexts. Future studies should conduct power analyses to determine appropriate sampling sizes. Second, the lack of quantitative risk assessment methods—such as Monte Carlo simulations and dose-response modelling—hinders the precise characterization of hazard severity and exposure thresholds. It is recommended that probabilistic risk modelling be incorporated to establish contamination benchmarks and prioritize intervention targets. A dietary intake risk assessment using toxicological and dietary data is needed for the chemicals in fruit and vegetable samples.

  1. Preface: Please remove the titles 1.1 and 1.2.
  2. Please specify the limit of quantitation (LOQ) for microbial and chemical hazards.
  3. 1. Sampling: More details are needed on the number of samples collected from each farm, including whether duplicates were taken.
  4. Lines 126-128: Are there references to soil microbial analysis methods? Given the soil's microbial diversity, is it appropriate to apply food biological analysis methods to the soil?
  5. Line 169: Please include the standard curve concentration, linear equation, and correlation coefficient.
  6. Figure 3: Add error bars to the figure.

Author Response

Author's reply to the review report was submitted as a PDF file.

Reviewer 2 Report

Comments and Suggestions for Authors

The study focused on fruits, vegetables, soil, and irrigation water collected from small farms in Northwest Portugal. The safety status of these agricultural products was elucidated by means of analyzing and testing the residues or contamination of chemical and biological hazards. Furthermore, the farmer's influence on food safety was evaluated. The paper is interesting, contains a substantial amount of data, and contributes to the enhancement of food safety for local small-scale farmers. However, the paper requires refinement in the following areas:

  1. The summary is deficient in quantitative information data and must be improved.
  2. Line 117, "for 1 minute at velocity 2." what is velocity 2?
  3. Why there are two LODs and LOQs in the supplementary tables S1 and S2, the authors need to clarify the description;.
  4. Further verification of the consistency of the data in the tables and the article is required because of the large amount of data in the article;.
  5. Lines 436-437, “also originate from these raw materials. low levels . ”, appears to be mispunctuated;.
  6. The term “risk assessment” is reflected in the title of the paper, but the results are not reflected in the main text; revise the title or improve the content;
  7. The analysis of farmers in the text lacks survey data, which should be added in the supplementary material.

Author Response

(The authors gave the same response as above.)

Reviewer 3 Report

Comments and Suggestions for Authors

1, the authors should state the number of samples collected from each farm.
2. Regarding the analytical method, the instrumental parameters should be listed in a table, and, it should be clear how LOD and LOQ are calculated and how the blank test is controlled.
3、The resolution of all the pictures needs to be improved.
4, The authors should clarify the factors for the selection of contaminant types, and, as there are many contaminants involved and some of them are detected in low quantities, the main description of substances and microorganisms with a high level of contamination can be given in the main text, and the rest of the text can be placed in the supplementary material.
5. The authors should analyse the comparative contamination of contaminants in agricultural products from foodstuffs originating from small farms versus those from large-scale plants with a high degree of industrialisation.

Author Response

(The authors gave the same response as above.)

Reviewer 4 Report

Comments and Suggestions for Authors

The scientific paper written by Macieira et al. presents a detailed analysis of the environmental and food risk assessment on pre-harvest activities from 17 local small-scale farms in the districts of Braga and Porto. The experimental design and the materials and methods section are well described, and the results are clearly defined. However, I suggest revise the paper to make it clearer for the readers and to ensure it has significant scientific relevance. Specifically:

Abstract:

It might be helpful if the authors could include more detailed results in the abstract. This would allow readers to immediately undestrand the key findings and the significance of the study.

Introduction:

About the introduction, the authors clearly explain the chemical risks associated with fruit and vegetable production but the microbiological risks are not well defined.

Line 42: in Section 1.1, the authors reported data on the microbiological risks associated with fruit and vegetable production. However, it may be beneficial to provide more clarity on the microbiological risks, especially in terms of transmission routes and concentrations of pathogenic bacteria in food items. Including this information could give a more comprehensive overview of the topic.

Line 43: The authors considered very few references about the detection of some pathogenic bacteria in food matrices. Please add other references.

Line 49: Assessing microbial safety risks is also related to the detection of enteric viruses in fruits and vegetables. Why did the authors not consider enteric viruses?

Line 50: The authors could explain what are the difficulties of eliminating contamination before and later in the food chain.

Line 54: (now banned): The phrase 'now banned' could be further clarified.

Line 71: Nitrate (NO₃⁻) instead of "NO3-" is more formal and precise.

Line 72: Please, add "European Food Safety Authority"

line 83: Please, delete "risks" before "these risks"

Results and Discussion:

While the authors have obtained significant and scientifically valuable results, I recommend a more integrated approach to connecting the findings from soil, water, and produce samples. Specifically, a more detailed explanation of the factors contributing to the observed prevalence of certain contaminants in specific sampling areas, as compared to others, would strengthen the overall interpretation of the manuscript. Furthermore, I suggest separating the results and discussion sections. This could make the paper easier to follow and allow for a more systematic presentation and analysis of the findings to facilitate a more coherent and structured presentation. By addressing the results in a more unified and systematic manner, the relevance and impact of the authors' scientific work would be more clearly communicated.

Author Response

(The authors gave the same response as above.)

Round 2

Reviewer 1 Report

Comments and Suggestions for Authors

The original title of the paper, which includes "risk assessment," remains appropriate.

The study revealed that concentrations of lead (Pb), cadmium (Cd), and zinc (Zn) in organic carrots (Daucus carota), beet leaves (Beta vulgaris), and chili peppers (Capsicum annuum) exceeded established regulatory thresholds. These findings underscore the need for further investigation into associated health risks for consumers.

Author Response

Comment: The study revealed that concentrations of lead (Pb), cadmium (Cd), and zinc (Zn) in organic carrots (Daucus carota), beet leaves (Beta vulgaris), and chili peppers (Capsicum annuum) exceeded established regulatory thresholds. These findings underscore the need for further investigation into associated health risks for consumers.

Response: We would like to thank you once again for your valuable insights. We appreciate your careful review and are pleased to note that the modification you suggested regarding heavy metals, has already been addressed and is now clearly highlighted in the conclusions section: “Although only a small number of food samples contained heavy metals such as Pb and Cd at levels exceeding regulatory thresholds, these occurrences highlight the importance of further research into the associated health risks for consumers.”

Reviewer 3 Report

Comments and Suggestions for Authors

The authors have revised the manuscript in response to the comments, and its quality has been significantly improved and it is recommended for acceptance.

Author Response

Comment: The authors have revised the manuscript in response to the comments, and its quality has been significantly improved and it is recommended for acceptance.

Response: We would like to express our sincere gratitude to you for your constructive feedback and for accepting the manuscript. Your thoughtful evaluations have contributed to improving the quality and clarity of our work.

Reviewer 4 Report

Comments and Suggestions for Authors

The authors have revised the manuscript, clarifying certain aspects and topics. I accept the manuscript in its current form.

Author Response

Comment: The authors have revised the manuscript, clarifying certain aspects and topics. I accept the manuscript in its current form.

Response: We would like to express our sincere gratitude to you for your constructive feedback and for accepting the manuscript. Your thoughtful evaluations have contributed to improving the quality and clarity of our work.